# Coincident onset of charge order and pseudogap in a homogeneous high-temperature superconductor

D. Betto[1,4], S. Nakata [2,4], F. Pisani [1], Y. Liu [2], S. Hameed[2], M. Knauft [2], C. T. Lin[2], R. Sant [1], K. Kummer [1], F. Yakhou[1], N. B. Brookes [1], M. Le Tacon [3], B. Keimer [2] & M. Minola [2] ✉

Understanding high-temperature superconductivity in cuprates requires knowledge of the metallic phase it evolves from, particularly the pseudogap profoundly affecting the electronic properties at low carrier densities. A key question is the influence of chemical disorder, which is ubiquitous but exceedingly difficult to model. Using resonant X-ray scattering, we identified two-dimensional charge order in stoichiometric $YBa_2Cu_4O_8$ ($T_c = 80$ K), which is nearly free of chemical disorder. The charge order amplitude shows a concave temperature dependence and vanishes sharply at $T^* = 200$ K, the onset of a prominent pseudogap previously determined by spectroscopy, suggesting a causal link between these phenomena. The gradual onset of charge order in other cuprates is thus likely attributable to an inhomogeneous distribution of charge ordering temperatures due to disorder induced by chemical substitution. The relationship between the pseudogap and the disorder-induced gradual freeze-out of charge carriers remains a central issue in research on high-$T_c$ superconductors.

The doping-driven transition from Mott insulator to high-temperature superconductor in copper oxides has become emblematic for quantum many-body physics[1], and the quest for a microscopic understanding of the underlying phase diagram has inspired countless advances in research areas ranging from materials exploration in twistronic heterostructures[2] and metal-oxide films[3] to quantum simulation with cold atoms[4] and solid-state nanostructures[5]. Despite these developments, the origin of one of the most prominent features of the cuprate phase diagram—the pseudogap phenomenon observed over a wide range of doping levels $p$ and temperatures $T$ (Fig. 1a)—remains unresolved. Recent evidence from transport and thermodynamics suggests that the pseudogap disappears via a quantum phase transition at a critical doping level $p^* = 0.19$ holes per planar unit cell[6]. While this observation implies a line of thermal phase transitions $T^*(p)$ emanating from $p^*$ (Fig. 1a), most experimental probes only indicate a

crossover (rather than a phase transition) at $T^*$. Notable exceptions are (partially controversial[7–9]) signatures of phase transitions in the uniform magnetization[10] and in a neutron scattering channel indicative of intra-unit-cell magnetic order[11]. However, the corresponding order parameters with wavevector $q = 0$ do not gap the Fermi surface, and can therefore not be the primary origins of the pseudogap phenomenon. Quasi-two-dimensional (quasi-2D) charge order with incommensurate $q \neq 0$ has recently been identified as the leading competitor of high-temperature superconductivity in moderately doped cuprates[12]. This form of order is known to induce partial gaps on the Fermi surfaces of compounds with quasi-2D electronic structure[13], and there are many indications for a Fermi surface reconstruction induced by charge order in the cuprates[6]. However, the amplitude of the charge-order parameter measured by X-ray diffraction only exhibits a gradual onset with temperature, and the characteristic onset

[1]European Synchrotron Radiation Facility (ESRF), BP 220, F-38043 Grenoble Cedex, France. [2]Max Planck Institute for Solid State Research, Heisenbergstraße 1, D-70569 Stuttgart, Germany. [3]Institute for Quantum Materials and Technologies, Karlsruhe Institute of Technology, Kaiserstr. 12, 76131 Karlsruhe, Germany. [4]These authors contributed equally: D. Betto, S. Nakata. ✉e-mail: m.minola@fkf.mpg.de

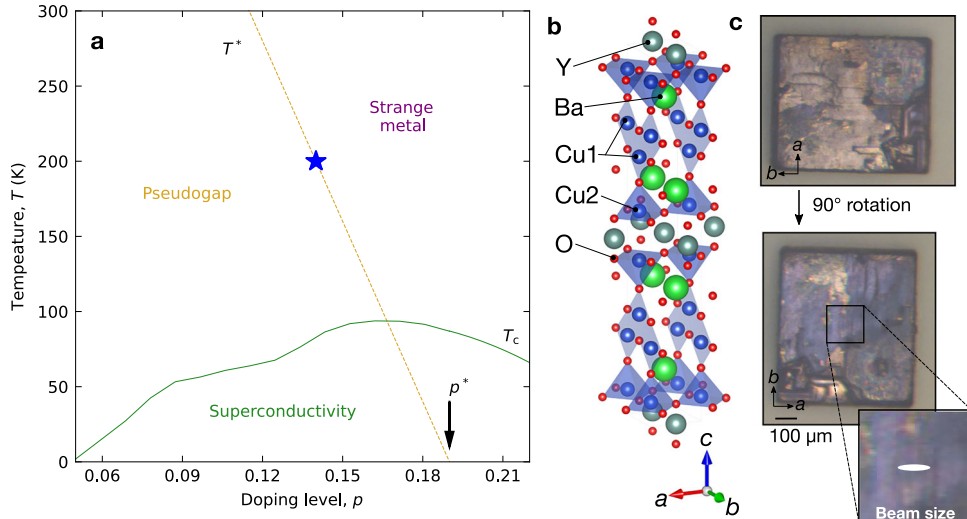

**Fig. 1 | RIXS from YBa₂Cu₄O₈. a** Temperature-doping phase diagram of YBa₂Cu₃O₆₊ₓ (Y123). The characteristic regimes (i.e., superconductivity, pseudogap, and strange metal) are indicated. The $T^*(p)$ line follows recent surveys of the experimental literature[1,6]. The blue star indicates the doping level and pseudogap temperature $T^*(p = 0.14)$ of YBa₂Cu₄O₈ (Y124) studied in the present work[19]. **b** Unit cell of YBa₂Cu₄O₈[69]. Cu1 and Cu2 indicate copper atoms in CuO chain layers and CuO₂ planes, respectively. **c** Sample surfaces imaged through a polarized microscope. The top and center panels display the same surface observed at different in-plane angles. The light (dark) colored regions in the top (center) panel indicate twin domains indicated by the crystallographic axes in the legend. The white oval in the bottom panel indicates the X-ray beam spot on the sample during the RIXS measurements.

temperature estimated from the X-ray signal is generally below $T^*$ (with some notable exceptions; see e.g., refs. 14,15 and Supplementary Note 6). Based on these considerations, the pseudogap and the $T^*(p)$ line in the phase diagram are widely regarded as unrelated to charge order[1,6,12].

Here we cast new light on this long-standing puzzle by highlighting the crucial influence of quenched disorder on the charge-ordering transition. Disorder is invariably associated with chemical substitution, the standard method of adjusting the doping level in the copper oxide planes. Since the combination of disorder and strong electronic correlations is an exceedingly difficult theoretical problem, most theoretical research on the cuprates targets hypothetical disorder-free systems. A look at other transition metal oxides, however, shows that substitutional disorder can profoundly affect the nature of charge-ordering transitions and the corresponding phase diagrams[16,17]. In manganates, for instance, chemical substitution has been observed to transform thermodynamic charge-ordering transitions into gradual freezing phenomena, thus obliterating the singularities in the macroscopic observables typically associated with phase transitions[16]. Moreover, general theoretical arguments imply that 2D systems with incommensurate charge order break up into finite-sized domains in response to arbitrarily low levels of disorder[18].

We have used resonant inelastic X-ray scattering (RIXS) to investigate the YBa₂Cu₄O₈ (Y124) system, which has played a prominent role in research on high-$T_c$ superconductivity because of its electronic homogeneity. Y124 is "self-doped" to $p = 0.14$ (corresponding to a superconducting $T_c$ of 80 K) via charge transfer from stoichiometric CuO chain units running along the crystallographic $b$ axis (Fig. 1b)[19]. Since Nuclear Quadrupole Resonance (NQR) provides a direct measure of the local doping level, the width of NQR lines is a potent signature of the spatial inhomogeneity of the doping level. The Cu NQR linewidths of stoichiometric Y124 (underdoped) and YBa₂Cu₃O₆₊ₓ (Y123) with $x = 1$ (slightly overdoped) are about an order of magnitude smaller than those of other cuprates with substitutional disorder, manifesting exceptional electronic homogeneity[20–22]. Prior experiments have shown that a pseudogap in Y124 opens up below $T^* \sim 200$ K, in line with most other cuprates at this doping level (Fig. 1a)[23–25]. We have discovered X-ray superstructure reflections indicative of charge order that exhibit a sharp onset at the same temperature, 200 K, in a manner

reminiscent of a second-order phase transition. The sharp, coincident onset of charge order and pseudogap in a minimally disordered system indicates a causal relationship between both phenomena, and provides a greatly improved basis for comparison to theoretical work on strongly correlated metals.

## Results and discussion

The RIXS measurements were carried out at the Cu $L$-edge in a polarization geometry that maximizes the cross section from charge correlations (see Methods). Figure 2a shows RIXS spectra taken at $T_c \sim 80$ K for various momenta along the $K$-axis in reciprocal space. (The Cartesian components of the momentum transfer ($H$, $K$, $L$) are given in reciprocal-lattice units; note that the $c$-axis parameters of Y124 and Y123 are different.) The inelastic part of the spectra comprises the spectrum of paramagnon excitations from −100 to −500 meV[26] and inter-orbital ($dd$) excitations from −1 to −3 eV[27]. These features do not show any clear intensity evolution versus momentum. Overall, both the energy and the momentum dependence of the RIXS spectra are very similar to those in Y123[28]. At zero energy transfer, the scattering signal exhibits a peak around $K \simeq -0.32$, which is highlighted in a momentum scan of the quasielastic intensity integrated over an energy window of ±100 meV (Fig. 2b). As reported for other cuprate families, this diffraction feature is a direct signature of incommensurate charge order. Nuclear Magnetic Resonance (NMR) experiments have confirmed that the charge correlations are static of a μsec time scale[29]. The scattering signal forms a weakly modulated rod along the out-of-plane direction (Fig. 2d), as expected for quasi-2D charge order. The in-plane scans were taken for $L \sim 3$, where the charge-order signal is maximal (see Supplementary Note 7).

Figure 2c displays the incident photon energy dependence of the quasielastic peak intensity, along with Cu $L_3$-edge X-ray absorption spectroscopy (XAS) data measured on the same crystal. The peak at 934.3 eV in the XAS profile is ascribed to the Cu1 atoms with $3d^{10}$ configuration located in the CuO chain layers (Fig. 1b)[30]. The electron configuration of the Cu2 atoms in the CuO₂ planes, on the other hand, is a superposition of $3d^9$ and $3d^9\underline{L}$ (with $\underline{L}$ indicating a ligand hole), with XAS peak energies at 931.3 and 932.8 eV, respectively. The quasielastic RIXS intensity is only present at the energy of the planar Cu2 atoms, and absent at the energy of the Cu1 atoms. This incident photon energy

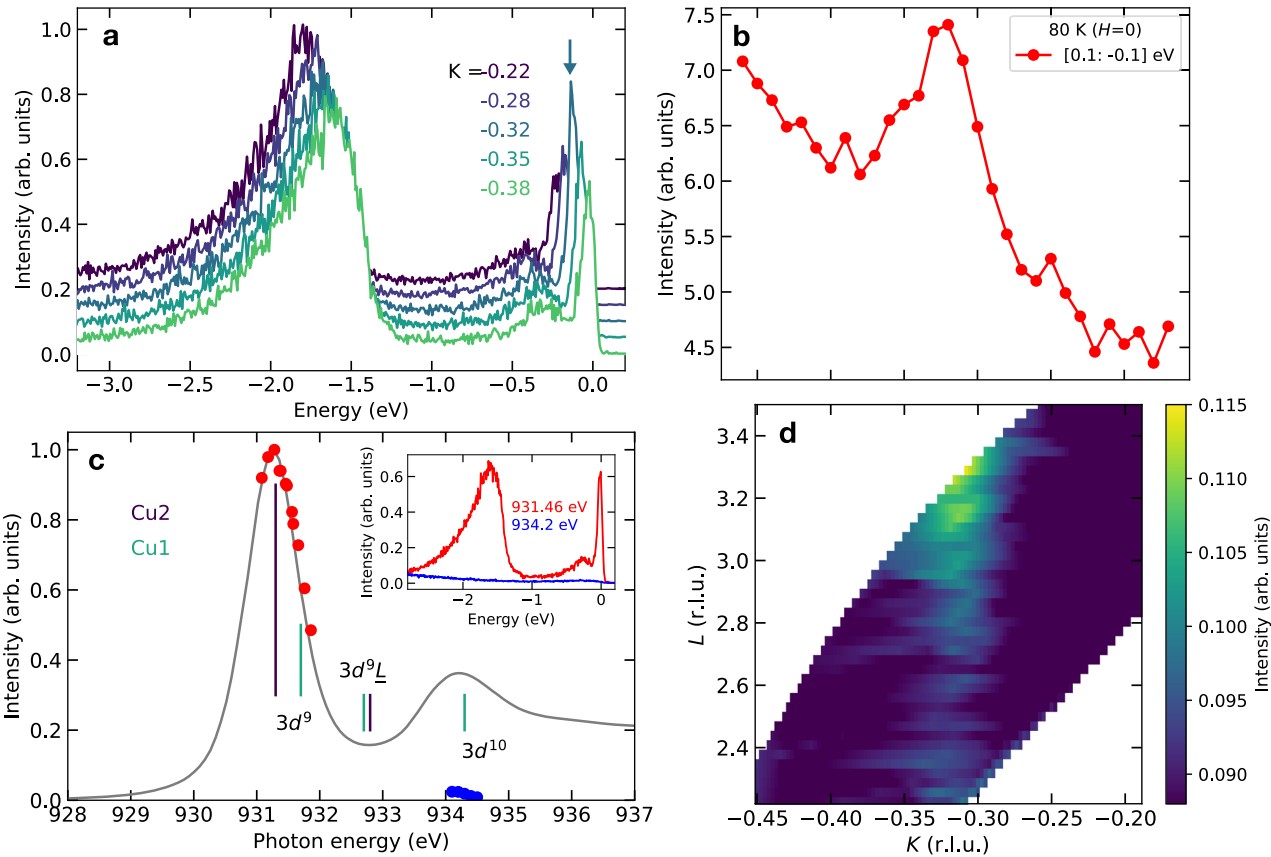

**Fig. 2 | RIXS and X-ray absorption spectra. a** RIXS spectra of Y124 taken at $T = 80$ K for various momenta along the $K$ axis. **b** Momentum dependence of the quasielastic intensity, defined as the RIXS intensity integrated in an energy window of width $\pm 100$ meV around zero energy loss. **c** XAS profile (gray line), and RIXS quasielastic intensity (red and blue points). The XAS and RIXS data were taken at room temperature and 80 K, respectively. Both XAS and RIXS intensities were normalized to

their maximum values in the measured energy range. The main absorption peaks of Cu1 and Cu2 for $3d^9$, $3d^9\underline{L}$, and $3d^{10}$ configurations are indicated by vertical bars[30]. Inset: RIXS spectra at $(H, K) = (0, -0.32)$ with incident photon energies of 931.46 eV (resonant, red) and 934.2 eV (off-resonant, blue). **d** Color map of the RIXS quasielastic intensity as a function of the momentum components $K$ and $L$.

dependence is also clearly apparent in the raw RIXS spectra at 931.46 eV and 934.2 eV (inset of Fig. 2c). These findings imply that the charge order is formed exclusively by valence electrons in the $CuO_2$ planes, with negligible contributions from the Cu1 atoms in the chains, analogous to prior results on Y123[31].

We now turn to the central result of our study, the temperature dependence of the superstructure peak intensity, which is proportional to the square of the amplitude of the electron-density modulation. Figure 3a shows momentum scans across these reflections at different temperatures. The intensity first increases upon heating up to the superconducting $T_c$, an effect that is also observed in Y123[28,32] and $La_{2-x}Sr_xCuO_4$[33] and can be attributed to a competition between charge order and superconductivity. In the normal state, the intensity then decreases continuously with increasing temperature, and for $T \geq 200$ K, only a featureless, temperature independent background is observed. Since the momentum width of the reflections is $T$-independent within the error, we fitted the data to Lorentzian profiles with fixed width and variable amplitude. Figure 3b shows the result of the least-squares fit. The concave $T$-dependence and sharp onset of the resonant superstructure reflections is reminiscent of a second-order phase transition in a clean system, and implies that static charge order sets in uniformly across the entire material. The onset of charge order in Y124 can thus be pinpointed to a sharply defined transition temperature.

Importantly, this temperature coincides with $T^* \sim 200$ K, where the opening of a pseudogap with amplitude ~50 meV has been

determined directly by infrared[34] and Raman[35] spectroscopy (Fig. 3c). Various other experimental observables also exhibit sharp anomalies at the same temperature and have thus served as proxies of the pseudogap, including the NMR Knight shift[23,25] shown in Fig. 3c, as well as the linewidth of Ho crystal-field levels in the isostructural compound $HoCu_2Cu_4O_8$ observed by neutron scattering[36]. Isotope effects reported for the latter two quantities[36,37] indicate strong coupling of the corresponding order parameter to the crystal lattice, prompting interpretations in terms of a charge-density wave (CDW)[15,38]. The direct observation of charge order now confirms these predictions. The temperature dependence of the resistivity has been interpreted in terms of a crossover with characteristic temperature around 200 K[24], although sharp anomalies are not observed at this temperature[39]. This is in line with the phenomenology in classical quasi-2D CDW metals such as $NbSe_2$, where only weak resistivity anomalies are observed at the critical temperature, because CDW formation gaps the Fermi surface only partially, and the influence of the loss in carrier density in the CDW state is partially compensated by the enhanced relaxation time of the remaining carriers[40].

It is interesting to compare the Y124 data to the gradual onset and convex temperature dependence of the charge order parameter other cuprates including Y123 (Fig. 3b), which had previously been interpreted in terms of an increased phase space of fluctuations between nearly degenerate quantum ground states[41,42]. With a minimally disordered cuprate as a baseline for comparison, this behavior can now be attributed to an inhomogeneous distribution of charge ordering

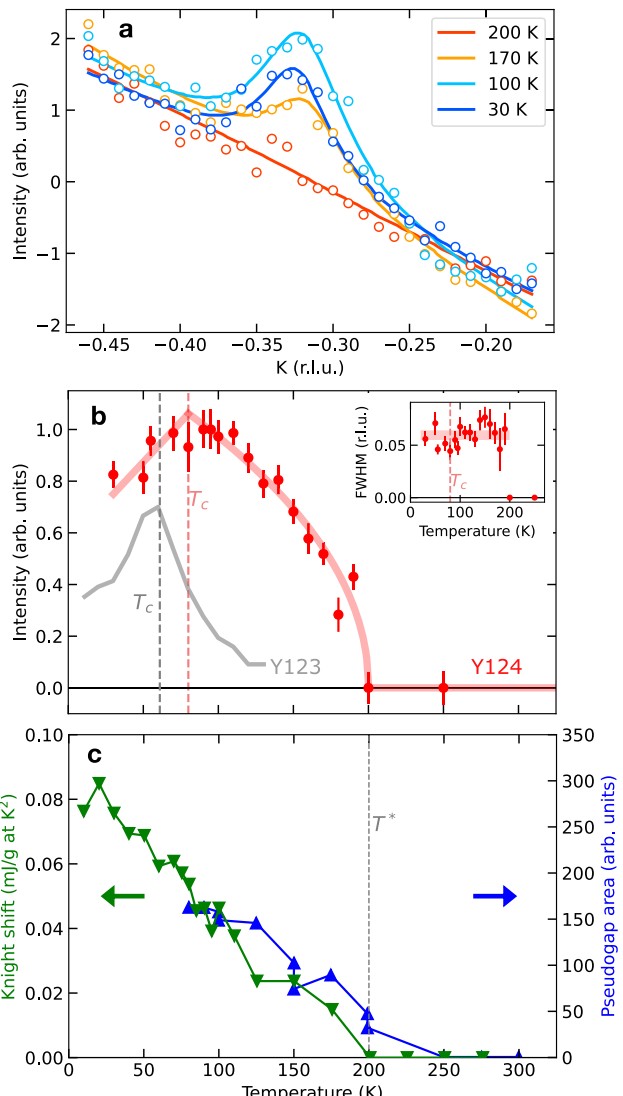

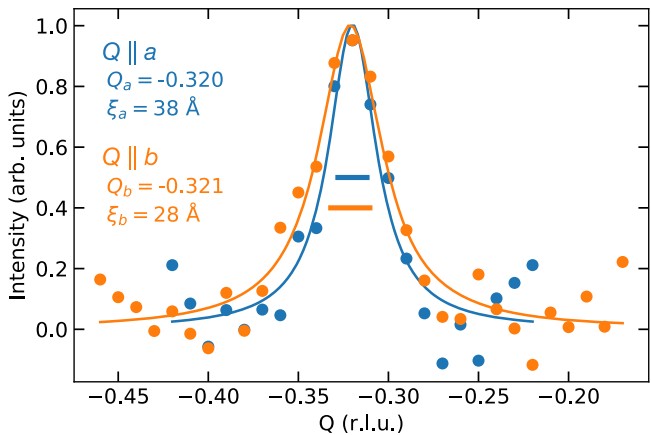

**Fig. 4 | Momentum dependence of the quasielastic intensity.** Quasielastic scans across $(H, K) = (0, -0.32)$ $((-0.32, 0))$ shown in orange (blue). For clarity, a linear background was subtracted from the raw quasielastic signal (Figs. 2c and 3a). The lines are the results of fits to Lorentzian profiles (see the main text for details). The horizontal bars denote the intrinsic momentum widths of the reflections.

**Fig. 3 | Temperature dependence of the charge order amplitude. a** Quasielastic intensity around $(0, -0.32)$ at selected temperatures (see Supplementary Note 3 for the data at other temperatures). The solid curves are results of least-squares fits to Lorentzian profiles with a linear background. **b** Temperature dependence of the quasielastic peak intensity based on the Lorentzian fits. The error bars represent the standard deviation of the fitting parameter. The red curve for $T_c \leq T \leq 200$ K is the result of a power-law fit with exponent $\beta = 0.24 \pm 0.07$. For comparison, the data of Y123 with doping level $p \sim 0.12$ are displayed as a gray curve, reproduced from ref. 45. Inset: Temperature dependence of the quasielastic peak intensity based on the Lorentzian fits. Details are described in Supplementary Note 5. **c** Anomalies in different observables associated with pseudogap formation. Green data points and left scale: Difference in O2 and O3 spin susceptibilities obtained from Knight shifts, $^{17}K_{2,c} - {}^{17}K_{3,c}$, in NMR measurements on Y124, reproduced from ref. 23. Blue data points and right scale: Amplitude of the pseudogap measured by electronic Raman scattering, reproduced from ref. 35.

temperatures due to disorder induced by chemical substitution, akin to glassy freezing phenomena in other perovskites with chemical disorder[16,17]. A closely parallel phenomenology is also observed in NbSe$_2$. Whereas the CDW transition in pristine NbSe$_2$ is marked by CDW Bragg reflections with an order-parameter-like onset and a clearly recognizable resistivity anomaly, both anomalies are weakened and drawn out over an extended temperature range when disorder is induced by electron irradiation[40]. Similarly, the charge ordering transition in the CDW material Er$_{1-x}$Pd$_x$Te$_3$ is smeared out by the disorder

introduced by increasing Pd doping[43]. Without prior knowledge about the clean system, their association would hence be difficult to recognize. Likewise, the gradual onset of the CDW signal in most cuprates makes it difficult to firmly establish correlations with the pseudogap. However, the pseudogap onset temperature is ~200 K for most cuprates with $p \sim 0.14$, consistent with the notion that the association we observed in Y124 is generic[1,44]. Since the doping level of Y124 is fixed and cannot be adjusted without introducing disorder, our data do not allow direct conclusions about the association between charge order and the pseudogap at other doping levels. We note, however, that the sharp thermal transition we observed is consistent with a line of phase transitions emanating from the pseudogap quantum critical point at $p^* \sim 0.19$ inferred from previous work (Fig. 1).

We finally address the characteristic size and shape of the charge-ordered domains inferred from the momentum distribution of the resonant scattering intensity. Figure 4 shows scans of the quasielastic intensities across $(H, K) = (0, -0.32)$ and $(-0.32, 0)$, after subtraction of the linear background signal (see Supplementary Note 3). The presence of both reflections indicates coexisting charge modulations propagating along the $a$- and $b$-directions with nearly identical periodicities within a single twin domain, mirroring the situation in untwinned YBa$_2$Cu$_3$O$_{6+x}$ with $x \sim 0.8$ and $p \sim 0.14$. The characteristic dimensions of charge-ordered domains (3–4 nm, i.e., 7–10 lattice spacings along $b$ and $a$, respectively) inferred from the longitudinal momentum width of the reflections, the elongated shape from the corresponding transverse in-plane scans (Supplementary Note 4), and the antiphase correlation of charge-ordered states in adjacent bilayers from scans perpendicuar to the CuO$_2$ layers (Fig. 1b) are also similar to the ones in Y123 at the same doping level[45].

The close analogy between Y124 and Y123 suggests that the domain shape is not strongly influenced by disorder—in marked contrast to the onset temperature. This conclusion is consistent with prior work on more strongly underdoped Y123[46], which found that the domain size is largely independent of the distribution of oxygen defects (the main source of disorder in this system). Charge order in other cuprates with different doping mechanisms also exhibits domain configurations with similar characteristic dimensions (see e.g., ref. 47), suggesting that they are primarily controlled by the intrinsic properties of the electron system in the CuO$_2$ planes, rather than materials-specific disorder. In addition to general theoretical arguments on the instability of incommensurate CDWs to domain formation[18], specific theoretical work indeed predicts polymer-like properties of uniaxially charge-ordered states in the CuO$_2$ planes, with a high susceptibility to

transverse fluctuations and topological defects[48–50]. The near-degeneracy of multiple forms of order in the strongly correlated electron system[51] further lowers the energy cost of domain walls, thus presumably contributing to the small length scale characterizing the domain pattern. Larger correlation lengths (~30 nm) have been observed in Y123 with $p \sim 1/8$ and in high magnetic fields, which weaken superconductivity and enhance the relative stability of charge order[52]. By analogy, the correlation length in Y124 is expected to grow in high magnetic fields, underscoring the potential of models based on charge order to explain the partial gap and Fermi surface reconstruction observed in quantum transport experiments[53,54].

In summary, we have discovered a direct diffraction signature of incommensurate, quasi-2D charge order in Y124, which had escaped detection despite the large body of experimental work on this system since its discovery in 1988. The reflections are hard to detect because their intensity is spread out over a ridge perpendicular to the $CuO_2$ planes and over a range of in-plane momenta corresponding to the finite domain size. Their observation was made possible by the synthesis of sizable Y124 single crystals, combined with the high signal-to-background ratio of energy-resolved resonant X-ray scattering and the precise control of all degrees of freedom allowed by the in-vacuum goniometer of the ERIXS spectrometer at the beamline ID32 in European Synchrotron Radiation Facility (ESRF). The sharp, coincident onset of charge order and the pseudogap at $T^* = 200$ K strongly suggests a causal relationship between both phenomena, for which there is a solid theoretical basis and ample precedent. In particular, the discovery of quasi-2D charge order in the stoichiometric metal $Sr_3Fe_2O_7$ by resonant X-ray scattering[55] has recently resolved the origin of transport and thermodynamic anomalies that had been enigmatic for many years, closely analogous to the situation in the cuprates.

These considerations suggest that inhomogeneous charge order plays a key role for the widely investigated onset of a pseudogap along the $T^*(p)$ line in Fig. 1a[1,6]. We note that thermodynamic data on Y124 indicate a second pseudogap with an onset at a much higher temperature[23], in line with the observation of a high-temperature "spin gap" controlled by the antiferromagnetic exchange coupling $J \sim 100$ meV in other cuprates (see e.g., refs. [56,57]). Our observation of paramagnon excitations (Fig. 2a) underscores the influence of exchange interactions on electronic correlations in Y124. Our results and the data of Ref. 23 are thus compatible with a scenario in which nearest-neighbor, dynamical spin-singlet correlations form well above room temperature, and $T^*$ marks their eventual condensation into a state with a static, bond-centered charge modulation[58–61]. Note that recent high-field NMR data on underdoped Y123 have also been interpreted in terms of a two-gap scenario with a high-energy spin gap that increases strongly with decreasing doping, and a low-energy, more weakly doping-dependent gap related to charge ordering[61]. Upon further cooling below $T_c$, superconductivity develops in Y124 on the backdrop of a charge-ordered texture with characteristic domain size comparable to the superconducting coherence length (~2 nm). As charge order is weakened but not obliterated by the onset of superconductivity (Fig. 3b), both forms of electronic order are intrinsically "intertwined" at this length scale. Y124 offers a unique platform to study this intertwined ground state with minimal interference from uncontrolled disorder. Our results open up numerous perspectives for further experimental and theoretical exploration, including the response of charge order to strain and magnetic fields, and the effect of controlled disorder on the microscopic charge correlations and macroscopic electronic properties.

Finally, we note that our results do not speak directly to recently reported observations of charge order in overdoped cuprates[62,63], where a pseudogap is not present but different factors such as a van-Hove singularity and oxygen-vacancy order[64] may be relevant.

## Methods

### Materials and sample characterization

High-quality Y124 single crystals with in-plane dimensions ~$500 \times 500$ μm² were grown by a flux method at ambient pressure[65,66] (Supplementary Note 1). The crystals contain twin domains with $a$- and $b$-axes interchanged. Prior to the RIXS measurements, we checked the surfaces of samples through a polarized microscope[67] and selected a crystal with large twin domains (Fig. 1c). The microscope image of the sample's backside largely matches the one of the front (Fig. 1c), which implies that the domain distribution in the bulk is closely similar to the one on the surface (Supplementary Note 2). As the area of the X-ray beam (~5 × 50 μm²) is considerably smaller than the largest twin domain, we were able to keep it focused on this domain throughout the RIXS measurements.

### RIXS measurements

The RIXS experiments were performed using the ERIXS spectrometer at the beamline ID32 of the European Synchrotron Radiation Facility (ESRF) at the Cu $L_3$ absorption edge (931.5 eV) with a combined energy resolution of 50 meV and momentum resolution of 0.004 Å$^{-1}$[68]. All data presented here were collected using $\sigma$-polarized incident X-rays to maximize the scattering cross section in the charge sector[28]. As a probe of weak elastic superstructure reflections originating from 2D charge order, RIXS has the advantage that it exclusively addresses the Cu valence electrons, and that it discriminates against inelastic scattering and thus enhances the signal-to-background ratio compared to the more widely used resonant diffraction without energy analysis.

## Data availability

Data that support the findings of this study are available from the corresponding authors upon request.

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

## Acknowledgements

We thank Marc-Henri Julien and Nigel Hussey for discussions. We acknowledge the European Synchrotron Radiation Facility (ESRF) for provision of synchrotron radiation facilities and support at the beamline ID32 under proposal numbers IH-HC-3699 and IH-HC-3714.

## Author contributions

D.B., S.N., F.P., Y.L., S.H., M.K., R.S., K.K., N.B.B., F.Y., and M.M. performed the RIXS measurements and discussed the results with M.L.T. S.N. analyzed the data. C.T.L. synthesized the single crystals. S.N., F.P., and Y.L. characterized the single crystals investigated in the present work. M.M. and B.K. supervised the project. S.N., B.K., and M.M. wrote the manuscript with inputs from all coauthors.

## Funding

## Competing interests

The authors declare no competing interests.
