## [Transparent Peer Review file · Nature Communications]

Coincident onset of charge order and pseudogap in a homogeneous high-temperature superconductor

Corresponding Author: Dr Matteo Minola

Version 0:

Reviewer comments:

Reviewer #1

(Remarks to the Author)

Betto and coauthors have measured, for the first time by using resonant inelastic X-ray scattering, a single crystal of $\text{YBa}_2\text{Cu}_4\text{O}_8$ (Y124). This compound is a cuprate high- T_c superconductor, distinguished by its unique doping mechanism that does not involve ionic substitution or the introduction of oxygen vacancies, which typically increase disorder within the unit cell. Their study on this minimally disordered crystal has led to the unprecedented detection of a quasi-static charge density wave (CDW) at an incommensurate wave vector of approximately 0.32 rlu., whose energy and temperature dependence has been measured. Two claims have stemmed from this discovery: Firstly, the results suggest a potential causal relationship between the CDW and pseudogap phenomena. This inference is drawn from the observation that the onset temperature of the CDW coincides with T^* , as measured in the same compound using various techniques, including transport and NMR. Secondly, the occurrence of the CDW in Y124 is proposed to be linked to its minimal disorder compared to other cuprates. In contrast to Y124, in these other compounds the authors claim that the disappearance of the CDW peak with temperature is less abrupt, and the CDW onset temperature is lower than T^* .

The data collected is of good quality, strongly corroborating the first detection of CDW in this compound. The motivation of the paper, which is intricately linked to the first and main claim is excellent. Indeed, achieving a deeper comprehension of the normal state characteristics of cuprates, particularly the interplay between various phases and orders in the underdoped region, is imperative for shedding light on the enigmatic phenomenology underlying the development of high- T_c superconductivity.

For the reasons outlined above, the paper and its findings are situated within a broader context of recent and fertile research, albeit lacking novelty. Specifically, the authors have overlooked a recent study [Science 373, 1506–1510 (2021)] where a similar conclusion was reached in $\text{YBa}_2\text{Cu}_3\text{O}_{7-\delta}$ (Y123). This oversight is somewhat surprising, especially considering that one of the authors of the present manuscript contributed a perspective on the aforementioned work [Science 373, 1438–1439 (2021)]. In Y123, as in Y124, the coincidence of the CDW onset temperature and T^* has been demonstrated, particularly at doping levels above $p=0.11$. Additionally, strain has been utilized as an additional degree of freedom to manipulate CDW and establish that the alignment of these two temperatures is not merely coincidental. Therefore, the findings presented in Y124 represent an important confirmation of this previous result, warranting in principle publication. However, it is imperative that these results are contextualized appropriately within the existing body of literature.

The argument behind the second claim of the paper, i.e., that the coincidence of TCDW and T^* is closely tied to the minimal disorder intrinsic to Y124, is instead highly speculative, upon careful examination of the presented data, giving rise to several concerns.

First of all, it is important to note that the coincidence of TCDW and T^* has also been observed in Y123, as previously mentioned [see in particular Figures 5 and S8 in Science 373, 1506–1510 (2021)].

Secondly, the comparison between Y124 and Y123 presented in the paper, particularly in Figure 3b, is not meaningful. The temperature dependence, measured recently using RIXS by the authors, is compared with older data acquired using energy-integrated resonant x-ray scattering. Due to the different sensitivities of the two instruments, direct comparisons of the

intensity and onset temperature of Y123 and Y124 are untenable. Notably, employing the same instrument as used in Ref. [29] to measure the temperature dependence of CDW in Y123, no CDW was detected in Y124 [Science, 337, 821 (2012)]. Furthermore, it is pertinent to mention that the enhanced sensitivity of the latest RIXS facilities, such as ID32 at ESRF and I21 at Diamond Light Source, enables the detection of CDW onset temperatures in various cuprates at significantly higher temperatures than those measured in the initial pioneering RIXS experiments nearly a decade ago.

As highlighted earlier, in the pioneering energy-integrated resonant x-ray scattering experiments, CDW was detected in Y124, in line with the doping dependence of Y123. This occurrence poses a challenge to the scenario proposed by the authors, wherein CDW, in the absence of disorder, should uniformly manifest across the entire material. Shouldn't this sharper and more defined behavior compared to Y123 give rise to a stronger intensity?

If the coincidence of TCDW and T^* is indeed unique to Y124, obscured by disorder in other cuprates, one would expect to observe distinct normal states in Y124, particularly a more pronounced and abrupt change in resistance at T^* , indicating the transition from linear-in-T resistance. However, transport data reveal strikingly similar $R(T)$ behaviors for Y123 and Y124. Both exhibit a gradual transition from linear-in-T behavior at high temperatures to a more quadratic behavior at lower temperatures.

As a result of the concerns raised, I respectfully disagree with the second claim presented in the paper, and I find that, overall, the manuscript does not currently meet the standard suitable for publication in Nature Communications. However, given the significance of the discovery of CDW in Y214, which suggests a potential close connection between the pseudogap and CDW phenomena, which is however now new, I am open to reconsidering my stance. I would encourage the authors to provide a more thorough contextualization of their results within the current research landscape and to engage in a robust discussion regarding the rationale behind the second claim. This could involve restructuring and softening the argument or incorporating additional data that strongly supports it.

Another minor question arises regarding the authors' mention of a featureless, temperature-independent background observed above 200 K. This observation implies that the authors do not observe the broader peak associated with charge fluctuations at high temperatures. Do the authors have an explanation for this experimental evidence? In Figure 3a, there are points deviating from linear behavior at q values close to q_{CDW} , although the inferred peak height is minimal. First of all, is this a tiny hint of high temperature charge fluctuations? Secondly, Could this suppression be an artifact of the measurements being conducted at negative values of momentum? Notably, the background in Figure 3a appears very steep, differing from observations at positive q values, where the charge fluctuation peak are commonly measured. This discrepancy might, in principle, hide the detection of the broad-in- q and weak peak associated with charge fluctuations. Clarification on these points would enhance the understanding of the experimental results presented in the manuscript.

Reviewer #2

(Remarks to the Author)

The authors report the discovery of charge density wave (CDW) order in the stoichiometric superconductor $YBa_2Cu_4O_8$. This is believed to be a disorder free material because the absence of disorder in the chains. Surprisingly, the CDW on-set in temperature is sharper than in other cuprate systems and occurs close to T^* . I think this is an important new result that should be published in Nature Communications. The results has implications for the relationship of charge order and pseudogap in cuprates. The manuscript is very well presented, and I do not have any major comments. I give some suggestions below:

- 1) The authors say argue that $YBa_2Cu_4O_8$ is intrinsically clean, compared to say $YBa_2Cu_3O_{6+x}$, because oxygen interstitials are not required to dope the CuO_2 planes. However, no quantitative data is provided to support this. Can the authors provide data (or refer to data) to show that $YBa_2Cu_4O_8$ is relatively clean. Data might include electronic mean free paths determined from transport or quantum oscillations.
- 2) The authors say that the CDW reflection is a ridge of scattering perpendicular to the CuO_2 planes and that measurement are made at $L \sim 1.5$ (i.e. approximately 1.5). Can the authors provide their best estimate of L (e.g. $L=1.45$) and state explicitly that measurements were made in fixed L mode (assuming this is the case). This would be helpful for others wishing to study Y124.
- 3) The authors mention CDW correlation lengths in the text, e.g. "larger correlation length have been observed in Y123". Please can the authors give values of the correlation length for Y124 and values (with reference) for the larger values in Y123 they are talking about.
- 4) The temperature dependence of the FWHM of the quasi-elastic peak is shown in the SI figure 6. I see no reason to hide this interesting plot in the SI. Please can the authors include this plot (or one of the correlation length versus temperature) in the main text.

Reviewer #3

(Remarks to the Author)

D. Betto et al. describes a Cu L-edge resonant inelastic X-ray scattering study on the stoichiometric $\text{YBa}_2\text{Cu}_3\text{O}_7$ (Y124, superconducting $T_c = 80$ K) which is free of most sources of chemical disorder. They discovered that the two-dimensional (2D) charge-order (CO) sets in sharply at $T^* = 200$ K coinciding with the onset of pseudogap (PG) suggesting a close relationship between the two phenomena. Based on the result, the authors conclude that the gradual onset of CO in other cuprates can be attributed to an inhomogeneous distribution of CO temperatures due to disorder induced by chemical substitution. The PG physics is intimately connected with superconductivity (SC) in cuprates however the origin of PG is still strongly debated. CO is known to induce Fermi surface reconstruction and partial gaps in cuprates. However, the order parameter of CO of many cuprates have gradual onset temperatures below PG onset temperature T^* . The current work on Y124 which is almost chemical disorder free sheds new light on the issue and demonstrates the coincidence between the two phenomena. The data analysis and presentation of figures are clear, however, I have some concerns with regards to the part of the main text:

- (1) Throughout the main text, the authors repetitively highlight the sharp coincident onset of CO and PG phenomena which is the major result of the manuscript. The authors mentioned there are solid theoretical proposals on the coincidence, however, I am expecting a more in-depth discussion on the implication of the finding. For instance, on the one hand, 2D CO in the current work shows a clear competition with SC below T_c , similar to Y123 systems. If CO and PG are closely connected to each other, can authors address the mutual relationship among CO, PG, and SC at least for Y124 compound? We also know that in some extremely overdoped $\text{Bi}_2\text{201}$ cuprates CO is re-entrant to the phase diagram [YY Peng et al., Nature materials, 2018] and there is no PG physics in the overdoped region, and likely the CO physics is very different to the one discovered here. Comments on the relevance of the work to the current study should be provided. Also, the discussion of the PG phenomena in the spin sector could be further elaborated. For instance, can RIXS provide some relevant information in terms of spin-singlet correlation? The discussion should be linked in connection to the relationship between PG and CO.
- (2) Secondly, previous NMR studies on a lot of cuprates suggest they detect more localised or static charge order rather than the dynamic ones in X-ray scattering. From the results presented in Y124, it seems that NMR and X-ray scattering probe the same facet of the CO phenomena. It would be very useful if the authors could address along the direction.
- (3) The authors mentioned that the onset temperature of CO is subject to chemical disorder, however, that of the PG seems independent and has a universal temperature dependence regardless the specific cuprate compounds. Could authors elaborate a bit more why that is the case? The authors have found that the CO domain size is largely independent of the distribution of oxygen defects unlike the onset temperature of CO. Again, it would be very beneficial to general audience by providing a qualitative explanation.
- (4) Extrapolating the current major result, that is, the coinciding onset temperatures of PG and CO, to the conclusion of the gradual onset of CO in many other cuprates is due to the disorder induced by chemical substitution is very indirect and a bit of stretch. I would be much satisfied if the current work comprises a parallel study on Y124 with the introduction of disorder, either by chemical doping or by electron irradiation. If the study could demonstrate the shift of the sharp onset temperature CO to a gradual one and differentiate from that of PG, it would be a more sophisticated study to convince the readers about the conclusion.
- (5) Finally, a much minor point about the explanation of the CO. A lot of text goes to explain the resonance effect of CO in CuO_2 planes instead of CuO chains. In my opinion the text could be easily relocated to Supplementary Information as this has been confirmed by previous works extensively.

Version 1:

Reviewer comments:

Reviewer #1

(Remarks to the Author)

In their original manuscript, Betto et al. convincingly demonstrated the coincidence of charge density wave (CDW) onset and pseudogap temperature in $\text{YBa}_2\text{Cu}_3\text{O}_7$ (Y124) using resonant inelastic X-ray scattering (RIXS). I previously noted that this finding is significant enough to merit publication in Nature Communications. However, I raised two main concerns.

First, regarding the presentation of the data, I observed that the authors did not cite or properly contextualize recent literature presenting analogous results. Second, concerning the interpretation of the data, I did not find the attribution of this coincidence primarily to the minimal disorder and crystallographic perfection of Y124 compared to other cuprates to be very convincing.

In the revised version, I appreciated the authors' shift in focus, as indicated already by the title, which emphasizes the electronic homogeneity of Y124. This perspective highlights the potential for more precise examination of the coincidence of CDW and pseudogap onset using this "platform", with less uncertainty than in any other cuprate compounds. I agree with the authors that this argument significantly strengthens the paper.

Regarding my first concern, the authors have now appropriately cited a relevant work as Ref. 14, which presents similar findings. In their response, they acknowledged the importance of this work, stating that the comprehensive results emerging from these experiments underline the necessity of studying the effects of controlled disorder and inhomogeneity on orders such as CDW and pseudogap, thereby enhancing their entwining. I believe this point is important and suggest that the authors give the proper relevance to these findings not only citing the reference in question, but discussing the relevance of this previous work in the context of their findings.

Based on these considerations, and provided that these minor changes are made, I strongly recommend the paper for

publication in Nature Communications.

As an additional comment regarding Figure 3b, I remain unconvinced by the authors' arguments. With the advances in experimental techniques, the RIXS setup currently used by the authors to measure Y124 exhibits significantly improved sensitivity and signal-to-noise ratio compared to the equipment available approximately 15 years ago, on which the gray curve, reproduced from Ref. [35], is based. However, I would like to emphasize that Figure 3b does not impact the main findings or interpretations of the paper.

Reviewer #2

(Remarks to the Author)

I am happy that the authors have addressed my comments and recommend publication subject to addressing issues raised by the other referees.

Reviewer #3

(Remarks to the Author)

Please see the review report in the attachment.

Version 2:

Reviewer comments:

Reviewer #1

(Remarks to the Author)

The authors have adequately addressed the comments raised by the reviewers, including mine, and I reaffirm my recommendation for publication.

Reviewer #3

(Remarks to the Author)

In the revised manuscript, the authors have addressed the issue/confusion on the two Pseudogaps and made a clearer statement. Alongside the explanation of other minor points, I am happy with the current form and recommend publication.

Reply to Reviewers for Nature Communications manuscript NCOMMS-23-60389-T

Coincident onset of charge order and pseudogap in a homogeneous high-temperature superconductor

by D. Betto, S. Nakata et al.,

We thank all three reviewers for their careful and overall very positive assessment and for their insightful comments, which have helped us a lot in improving our manuscript. We very much welcome the opportunity to respond to the reviewers' comments, which are reproduced in blue.

Best regards,

Suguru Nakata, Matteo Minola, and Bernhard Keimer, on the behalf of all co-authors

Reviewer #1 (Remarks to the Author):

Betto and coauthors have measured, for the first time by using resonant inelastic X-ray scattering, a single crystal of YBa₂Cu₄O₈ (Y124). This compound is a cuprate high-T_c superconductor, distinguished by its unique doping mechanism that does not involve ionic substitution or the introduction of oxygen vacancies, which typically increase disorder within the unit cell. Their study on this minimally disordered crystal has led to the unprecedented detection of a quasi-static charge density wave (CDW) at an incommensurate wave vector of approximately 0.32 rlu., whose energy and temperature dependence has been measured. Two claims have stemmed from this discovery: Firstly, the results suggest a potential causal relationship between the CDW and pseudogap phenomena. This inference is drawn from the observation that the onset temperature of the CDW coincides with T*, as measured in the same compound using various techniques, including transport and NMR. Secondly, the occurrence of the CDW in Y124 is proposed to be linked to its minimal disorder compared to other cuprates. In contrast to Y124, in these other compounds the authors claim that the disappearance of the CDW peak with temperature is less abrupt, and the CDW onset temperature is lower than T*.

The data collected is of good quality, strongly corroborating the first detection of CDW in this compound. The motivation of the paper, which is intricately linked to the first and main claim is excellent. Indeed, achieving a deeper comprehension of the normal state characteristics of cuprates, particularly the interplay between various phases and orders in the underdoped region, is imperative for shedding light on the enigmatic phenomenology underlying the development of high-T_c superconductivity.

Our reply: We thank the reviewer for recognizing and appreciating the importance of our work and the quality of our data. Thanks to his/her criticisms we have now significantly improved our presentation. Below we address his/her concerns.

For the reasons outlined above, the paper and its findings are situated within a broader context of recent and fertile research, albeit lacking novelty. Specifically, the authors have overlooked a recent study [Science 373, 1506–1510 (2021)] where a similar conclusion was reached in YBa₂Cu₃O_{7- δ} (Y123). This oversight is somewhat surprising, especially considering that one of the authors of the present manuscript contributed a perspective on the aforementioned work [Science 373, 1438-1439 (2021)]. In Y123, as in Y124, the coincidence of the CDW onset temperature and T* has been demonstrated, particularly at doping levels above p=0.11. Additionally, strain has been utilized as an additional degree of freedom to manipulate CDW and establish that the alignment of these two temperatures is not merely coincidental. Therefore, the findings presented in Y124 represent an important confirmation of this previous result, warranting in principle publication. However, it is imperative that these results are contextualized appropriately within the existing body of literature.

Our reply: We thank the reviewer for pointing out this relevant prior study of Y123 thin films we had indeed overlooked. As correctly stated by the reviewer, this study yielded evidence of a coincidence of the CDW onset and the pseudogap onset temperature T*, contrary to the widespread notion that both

temperatures are different, and in agreement with our conclusion. To provide the appropriate context, we follow the reviewer’s suggestion and cite this study early on, at the end of the first paragraph (Ref. 14).

Following the reviewer’s comment, and a comment by reviewer #2 (below), we realized that we had not properly highlighted another essential part of the scientific context in the first version of our manuscript – namely the special role Y124 has played in research on high-temperature superconductivity. Whereas we had focused on its crystallographic perfection and lack of chemical disorder, the importance of Y124 derives from its **electronic homogeneity**. According to a comprehensive body of experimental (now cited as Refs. 21,22) and theoretical research (*e.g.* Chen *et al.*, now cited as Ref. 23), nuclear quadrupole resonance (NQR) provides a direct measure of the local doping level, and the width of the NQR lines is hence a signature of the spatial inhomogeneity of the doping level. Figure R1, which was copied from recent review article (now cited as Ref. 22) demonstrates that the Cu NQR linewidths of the stoichiometric compounds YBa₂Cu₄O₈ (underdoped) and YBa₂Cu₃O₇ (slightly overdoped) are at least an order of magnitude smaller than those of other compounds with substitutional disorder, including oxygen-deficient YBa₂Cu₃O_{7-δ}. A quantitative analysis (Ref. 23) has shown that the NQR linewidth for La_{1.94}Sr_{0.16}CuO₄ translates into a standard variation $\Delta p \sim 12\%$ of the local doping level, comparable to the average $p \sim 16\%$. The order-of-magnitude lower electronic inhomogeneity $\Delta p/p$ of YBa₂Cu₄O₈ compared to other underdoped cuprates observed by NQR is therefore highly significant, and it would be unreasonable not correlate this finding with the much sharper charge ordering transition we have discovered.

To emphasize homogeneity as a key quantitative characteristic of the electron system in YBa₂Cu₄O₈, we have added a brief paragraph to the introduction and modified the title of our paper:

*Coincident onset of charge order and pseudogap in a **homogeneous** ~~minimally disordered~~ high-temperature superconductor.*

Fig. R1. Planar Cu NQR spectra of various cuprate superconductors. Note that the linewidths of the stoichiometric compounds YBa₂Cu₄O₈ and YBa₂Cu₃O₇ and are an order of magnitude lower than those of all other compounds (Fig. 8.9 by Jurkutat *et al.*, now cited as Ref. 22, in A. Bussmann-Holder *et al.* (eds.), *High-T_c Copper Oxide Superconductors and Related Novel Materials*, Springer Series in Materials Science **255**, p. 77-96 (2017)).

Because of its special significance in research on high- T_c superconductivity, Y124 has been extensively characterized by a wide variety of experimental methods, including transport, NQR, NMR, infrared spectroscopy, and electronic Raman scattering. Note that the latter two methods allow direct spectroscopic determination of the pseudogap and its temperature dependence. All of these methods consistently indicate $T^* \sim 200$ K as the onset of the pseudogap. To highlight this important aspect of the scientific context, we have added the T-dependent amplitude of the pseudogap determined by electronic Raman scattering to Fig. 4; the new version of this figure is reproduced as Fig. R2 below. The coincidence of two sharply defined temperatures, T^* and T_{CDW} , is obvious from the experimental data, without any assumptions or fitting procedures.

Many other studies of the pseudogap rely on the T-dependent resistivity to estimate T^* . Specifically, the normal-state resistivity is fitted to a line, and the temperature at which the data begins to deviate from T-linear behavior is identified with T^* . As shown in Fig. 4 in the manuscript (left side of Fig. R2 below), the result of this procedure for Y124 is consistent with the determination of T^* by NMR and electronic Raman scattering. However, the determination depends significantly on the criterion set for the deviation from

linearity and on the temperature range available for linear fitting, and is hence associated with much greater uncertainty.

We therefore disagree with reviewer #1's statement that our results "lack novelty" in light of the earlier study (now cited as Ref. 14). To facilitate discussion, we have reproduced the relevant figures from Ref. 14 on the right side of Fig. R2. Note that the study was carried out on thin films, where NMR, NQR, IR and Raman measurements were not reported, and the pseudogap was determined based on resistivity measurements alone. The T-dependent CDW order is shown for one sample (a 10 nm thick film; blue stars in the upper panel), and for this sample, the temperature range with T-linear resistivity extends only from ~230 to 300 K (shaded range in the lower panels). We also note that the earlier study was carried out on underdoped Y123, which according to Fig. R1 is much less homogeneous than Y124, and that the charge redistribution at the surface and substrate interface of thin films adds another source of inhomogeneity. [As an aside, we note that a four-unit-cell (~4.5 nm) thick buffer layer with reduced doping level was recently found in Dy123 thin films and attributed to missing CuO chains at the substrate interface (Dawson *et al.*, PRL **125**, 237001 (2020))]. The study of Ref. 14 involved measurements on samples with different substrates and thicknesses, and the authors clearly deserve credit for spotting a parallel trend of the strain dependence of T^* and T_{CDW} despite all of these difficulties. Clearly, however, this study should not detract from the novelty of our results on a uniquely homogeneous and extensively characterized bulk compound with sharply defined T^* and T_{CDW} . In our view, it would be unreasonable to use this study as an argument to preclude publication of our results in Nature Communications.

Fig. R2. left: Fig. 4b,c in the main text of the manuscript, with the CDW order parameter (top) and various measures of the pseudogap (bottom) in Y124. We have added the T-dependent amplitude of the pseudogap extracted from electronic Raman scattering (Ref. 36) to panel c. right: CDW order parameter (top) and in-plane resistivity (bottom) of Y123, reproduced from Fig. S8 and Fig. 3A,B of Ref. 14, respectively.

The argument behind the second claim of the paper, i.e., that the coincidence of T_{CDW} and T^* is closely tied to the minimal disorder intrinsic to Y124, is instead highly speculative, upon careful examination of the presented data, giving rise to several concerns.

We wish to carefully discuss possible implications of our experimental discovery, and to refrain from any kind of speculative claim. This comment of reviewer #1 (and a related comment by reviewer #3 below) indicate that the original manuscript did not differentiate clearly enough between conclusions that can be drawn with a high degree of confidence, and open questions raised by our discovery which require further

research. In the following, we briefly summarize our two central conclusions and the main open question, and then review how we have revised the text in order to explain them more clearly.

Correlation between CDW order and pseudogap in Y124. Following the detailed discussion above, it is clear that both T^* and T_{CDW} in Y124 are sharply defined within a margin of error $\Delta T \sim 25$ K (Fig. R2), and that they coincide within the same margin (Fig. R2). It is thus implausible that both phenomena are unrelated. Indeed, a causal relationship is consistent with generally accepted experimental and theoretical work on materials with quasi-two-dimensional electron systems, according to which CDW order induces a Fermi surface reconstruction and a partial gap on the Fermi surface. This conclusion was accepted by all reviewers, and in our view, already justifies publication of our manuscript in Nature Communications.

Correlation between electronic homogeneity and T-dependence of CDW order parameter. Based on the arguments laid out above, there is also a strong case for a causal relationship between the sharp onset and concave T-dependence of the CDW order parameter in Y124 (which are characteristic of a second-order phase transition in a clean system) on the one hand, and the uniquely homogeneous electron system of this compound on the other hand. The much larger inhomogeneity of other underdoped cuprates (Fig. R1) provides by far the most plausible explanation for the convex T-dependence with rounded onset of CDW order in these systems, especially since closely similar observations have been reported for other CDW systems such as NbSe₂ (Ref. 41). We are aware of only a single alternative explanation for this behavior that is not based on disorder – namely the large phase space for fluctuations between nearly degenerate quantum ground states, which can be described by the SU(2) symmetry group (see the newly inserted references 42 and 43). However, this alternative scenario is generic to the cuprates and should therefore also apply to Y124. Based on our observation of concave T-dependence and sharp onset of CDW order in Y124, it can hence be dismissed. We are not aware of other possible scenarios that could explain the striking difference in T-dependence and sharpness of the CDW order parameter in compounds with different degrees of homogeneity.

Nonetheless, we have decided to soften our statement of the second conclusion in the abstract:

*Based on our results, the gradual onset of charge order in other cuprates **are likely attributable** to an inhomogeneous distribution of charge ordering temperatures due to disorder induced by chemical substitution.*

To emphasize the unique functional T-dependence of the CDW order parameter in Y124, we have reworded the following sentence in the abstract:

Remarkably, the charge-order parameter exhibits the concave temperature dependence characteristic of a second-order phase transition in a clean system and vanishes sharply at $T^ = 200$ K, which coincides with the onset of the pseudogap that was previously determined by spectroscopy, thus suggesting a causal relationship between the two phenomena.*

For completeness and in compliance with the reviewer's request for contextualization, we have added a brief discussion of the SU(2) scenario on page 4 and inserted two pertinent references [Hayward *et al.*, Science **343**, 1336 (2014); Kloss *et al.*, Rep. Prog. Phys. **79**, 084507 (2016)].

Open question: How do disorder and inhomogeneity influence CDW order and the pseudogap? Our findings on a system with minimal disorder and inhomogeneity open up a new and surprising perspective on the relationship between CDW order, the pseudogap, and doping-induced disorder, but we have absolutely no intention of pretending that they completely resolve this complex, long-standing issue. We clearly portray this relationship as an open problem in the abstract:

The relationship between the pseudogap and the gradual freeze-out of charge carriers in the presence of quenched disorder is a central issue in the description of the normal state of high-temperature superconductors.

and state the following caveat on page 4 of the original manuscript:

Since the doping level of Y124 is fixed and cannot be adjusted without introducing disorder, our data do not allow direct conclusions about the association between charge order and the pseudogap at other doping levels.

We still believe that these statements are adequate, but of course we are open to alternative wording suggestions.

We found that the following sentences on page 4 of the original manuscript were perhaps not optimally worded and might have contributed to the reviewer's concern:

In the cuprates, disorder-induced inhomogeneity likewise obscures the correspondence between charge order on the one hand, and anomalies associated with the pseudogap on the other hand. However, the pseudogap onset temperature is generally ~ 200 K for cuprates with $p \sim 0.14$, strongly suggesting that the association we observed in minimally disordered Y124 is generic.

We have rephrased these sentences as follows, and hope that they are now acceptable to the reviewer; if not, we would be grateful for alternative wording suggestions.

Likewise, the gradual onset of the CDW signal in most cuprates makes it difficult to firmly establish correlations with the pseudogap. However, the pseudogap onset temperature is generally ~ 200 K for cuprates with $p \sim 0.14$, consistent with the notion that the association we observed in Y124 is generic.

First of all, it is important to note that the coincidence of T_{CDW} and T^* has also been observed in Y123, as previously mentioned [see in particular Figures 5 and S8 in Science 373, 1506–1510 (2021)].

We refer to the discussion above. Our observation of coincident T^* and T_{CDW} in bulk crystals of Y124 can be stated with a high degree of confidence (conclusion 1 above). Disorder broadens the CDW transition (conclusion 2) so that it becomes harder to investigate the relationship between both phenomena; details depend on the disorder realized in the specific system under study, the way in which the pseudogap is determined, etc. The results of the Y123 thin film study mentioned by the reviewer are complementary to, and consistent with these conclusions, although they clearly do not reach the same level of confidence in the numerical identification of T^* and T_{CDW} . Both studies together motivate a systematic investigation of the influence of tightly controlled disorder and inhomogeneity on various experimental signatures of CDW order and the pseudogap.

Secondly, the comparison between Y124 and Y123 presented in the paper, particularly in Figure 3b, is not meaningful. The temperature dependence, measured recently using RIXS by the authors, is compared with older data acquired using energy-integrated resonant x-ray scattering. Due to the different sensitivities of the two instruments, direct comparisons of the intensity and onset temperature of Y123 and Y124 are untenable. Notably, employing the same instrument as used in Ref. [29] to measure the temperature dependence of CDW in Y123, no CDW was detected in Y124 [Science, 337, 821 (2012)]. Furthermore, it is pertinent to mention that the enhanced sensitivity of the latest RIXS facilities, such as ID32 at ESRF and I21 at Diamond Light Source, enables the detection of CDW onset temperatures in various cuprates at significantly higher temperatures than those measured in the initial pioneering RIXS experiments nearly a decade ago.

We disagree with the reviewer's statements in this paragraph.

First, the T-dependences of the CDW signal measured with different instruments are identical within the experimental error. This was already demonstrated in Fig. 4 of the paper by Ghiringhelli *et al.* (Science 337, 821 (2012)), which shows data on the same $YBa_2Cu_3O_{6.6}$ crystal taken at an energy-integrating REXS instrument and at a RIXS instrument with energy resolution ~ 130 meV. We have reproduced these data in Fig. R3 below, along with another data set taken on an instrument with even higher energy resolution (3 meV). Later work showed that the x-ray data are also identical to NMR measurements, which are sensitive to charge correlations in an even narrower energy window ($\sim \mu$ eV). It is therefore incorrect to say that data taken on different instruments cannot be meaningfully compared.

Fig. R3. (left) Temperature dependent intensity of the incommensurate CDW diffraction signal in identical $\text{YBa}_2\text{Cu}_3\text{O}_{6.6}$ crystals ($T_c = 61$ K) at different instruments with different energy discrimination: conventional energy-integrating REXS measured at beamline UE46-PGM1 at BESSY-II (black circles); RIXS measured with SAXES spectrometer at the ADDRESS beamline at the Swiss Light Source in an energy window of width $\Delta E = 130$ meV around $E=0$ (open circles); and high-resolution non-resonant x-ray scattering at beamline ID28 at the ESRF in an energy window of width $\Delta E = 3$ meV around $E=0$ (orange circles). Data from [10.1126/science.1223532] and [10.1038/nphys2805]. (right) Comparison of the onset of CDW signals in two sets of underdoped $\text{YBa}_2\text{Cu}_3\text{O}_{6+x}$ crystals detected by x-ray diffraction (green symbols); nuclear magnetic resonance (red symbols) and the polar Kerr effect (crosses). Figure from [10.1038/ncomms7438].

Second, the reviewer noted that the paper Ghiringhelli *et al.* (Science 337, 821 (2012)) reported a null result for an attempt to detect a CDW signal in Y124. These data were not taken by energy-integrating REXS (as the reviewer seems to imply), but at the SAXES spectrometer at the Swiss Light Source at Paul Scherrer Institute between May 2009 and 2011, with an inaccurate cold-finger manipulator that was not designed for fine diffraction scans. Moreover, the single-crystal samples available at that time were much smaller than those that were synthesized for the present study. In the present work, the full precision control of all degrees of freedom made possible by the in-vacuum diffractometer manipulator of ERIXS at ID32 at the ESRF, and the smaller beamspot ($\sim 5 \times 50 \mu\text{m}$) combined with the much larger samples ($\geq 500 \mu\text{m}$) allowed us to perform accurate measurement on a single, well aligned twin domain. The positive detection in the current study was thus enabled by technical progress in spectrometer design and sample synthesis. The failed attempt under much less favorable conditions more than a dozen years ago has not bearing whatsoever on the interpretation of our data.

Third, it may be correct that modern RIXS facilities with high background discrimination allow one to trace the CDW signal in Y123 and other cuprates with gradual onset to a higher temperature. Depending on what criterion one uses to define the onset temperature, this could possibly reduce the apparent difference between T^* and T_{CDW} inferred from prior work. However, this is not relevant for any of our conclusions, which are more clearly stated in the revised manuscript. The qualitative difference between the T-dependence of charge order in Y124 (concave with sharp onset) and Y123 (convex with gradual onset) is independent of measurement accuracy.

As highlighted earlier, in the pioneering energy-integrated resonant x-ray scattering experiments, CDW was detected in Y124, in line with the doping dependence of Y123. This occurrence poses a challenge to the scenario proposed by the authors, wherein CDW, in the absence of disorder, should uniformly manifest

across the entire material. Shouldn't this sharper and more defined behavior compared to Y123 give rise to a stronger intensity?

We surmise that a “not” is missing in the reviewer’s first sentence: “... CDW was *not* detected in Y124 ...”

This remark seems to be similar to the second point in the previous paragraph of the Reviewer’s report. Beyond the technical issues discussed there, we note that comparison of the absolute intensities of Y124 and Y123 samples would require carefully calibrated reference measurements under the exact same conditions, which are not available at this time. Moreover, one would have to compare both compounds at the same doping level. As the intensity of Y123 samples with the same doping level ($p \sim 0.14$) as Y124 are considerably smaller than those for $p \sim 1/8$, where most measurements were taken (see for example Blanco-Canosa *et al.*, Ref. 35). It is thus entirely possible that a quantitative comparison (which must be left for future work) will reveal a higher absolute intensity for Y124.

If the coincidence of TCDW and T^* is indeed unique to Y124, obscured by disorder in other cuprates, one would expect to observe distinct normal states in Y124, particularly a more pronounced and abrupt change in resistance at T^* , indicating the transition from linear-in- T resistance. However, transport data reveal strikingly similar $R(T)$ behaviors for Y123 and Y124. Both exhibit a gradual transition from linear-in- T behavior at high temperatures to a more quadratic behavior at lower temperatures.

We thank the reviewer for making this interesting point. A priori, one might indeed have expected a more pronounced anomaly of the resistivity at the CDW transition of Y124, and we cannot claim to fully understand why only a crossover is found. It turns out, however, that many materials with quasi-two-dimensional electron system only exhibit very weak resistivity anomalies at T_{CDW} , because the Fermi surface is only partially gapped in the CDW state, and the influence of the loss in carrier density in the CDW state is partially compensated by the enhanced relaxation time of the remaining carriers. We have inserted a corresponding remark on page 4 of our manuscript. Figure R4 shows two examples of resistivity measurements on the classical CDW compound NbSe₂, which has a simple lattice structure and is perfectly stoichiometric. One of these studies (Fig. R4, right panels) shows that the weak resistivity anomaly at T_{CDW} is completely obliterated by a small density of defects generated by electron illumination. As our Y124 crystals were grown in Al₂O₃ crucibles, we cannot rule out a small density of Al point defects that may adversely affect the transport properties. Prior work on the Y123 system has shown that growth in BaZrO₃ crucibles can greatly enhance the transport mean free paths; an analogous synthesis effort in our laboratory will focus on enhancing the purity of our crystals specifically for transport measurements.

Fig. R4. Examples of resistivity measurements on NbSe₂, which exhibits a charge density wave transition at $T_{CDW} = 33$ K. The figures are reproduced from (left) Corcoran *et al.*, *J. Phys.: Condens. Matter* **6**, 4419-4492 (1994) and (right) Cho *et al.*, *Nature Comm.* **9**, 2796 (2018). The figure on the right shows that the weak resistivity anomaly at T_{CDW} is obliterated by a small density of defects generated by electron irradiation.

As a result of the concerns raised, I respectfully disagree with the second claim presented in the paper, and I find that, overall, the manuscript does not currently meet the standard suitable for publication in Nature Communications. However, given the significance of the discovery of CDW in Y214, which suggests a

potential close connection between the pseudogap and CDW phenomena, which is however now new, I am open to reconsidering my stance. I would encourage the authors to provide a more thorough contextualization of their results within the current research landscape and to engage in a robust discussion regarding the rationale behind the second claim. This could involve restructuring and softening the argument or incorporating additional data that strongly supports it.

Our reply: Following the reviewer's insightful comments, we have softened the language in the abstract, sharpened the statement of our conclusions, enhanced the discussion of the scientific context, added several new references, and responded in detail to the Reviewer's concerns. We are confident that he/she will find the new version of the manuscript suitable for publication.

Another minor question arises regarding the authors' mention of a featureless, temperature-independent background observed above 200 K. This observation implies that the authors do not observe the broader peak associated with charge fluctuations at high temperatures. Do the authors have an explanation for this experimental evidence? In Figure 3a, there are points deviating from linear behavior at q values close to q_{CDW} , although the inferred peak height is minimal. First of all, is this a tiny hint of high temperature charge fluctuations? Secondly, could this suppression be an artifact of the measurements being conducted at negative values of momentum? Notably, the background in Figure 3a appears very steep, differing from observations at positive q values, where the charge fluctuation peak are commonly measured. This discrepancy might, in principle, hide the detection of the broad-in- q and weak peak associated with charge fluctuations. Clarification on these points would enhance the understanding of the experimental results presented in the manuscript.

Our reply: As explained in the main text, we did not find any temperature-dependent broader peak that could be ascribed to charge density fluctuations similar to those in Fig. 2 of Science **365**, 906-910 (2019) by Arpaia *et al.* The background was nearly linear and identical at all high temperatures and every time we measured it. No statistically significant sign of a broad peak was detected and we believe there was no hint of clear charge density fluctuations, although we were measuring with the very same setup used for Science **365**, 906-910 (2019). The two points not lying directly on the linear background fit noticed by the reviewer cannot really be distinguished from the noise.

We respectfully disagree with the reviewer about the optimal q -space location to measure charge order and fluctuations. Some of us have been involved in many published CDW experiments and since the very beginning the negative- q side (near grazing incidence, near normal outgoing) was the location of choice for charge order studies. The positive- q side (with π incident polarization) has been the preferred choice for paramagnons, because the near-grazing outgoing direction suppresses π -polarized emission and thus favors π -sigma cross-polarized magnetic scattering. Most resonant scattering experiments also explored the positive side and always found symmetric results regarding the CDW. In support of the idea that the negative- q location allows the detection of weak or broad charge order signals, it suffices to look at the data in Phys. Rev. B **98**, 161114(R) (2018) where some of us, using again the very same RIXS setup, easily measured weak elastic and inelastic charge order signals, also at high temperatures, in electron-doped cuprates.

At this stage, it is too early to conclude that the broad peak Arpaia and other observed in Y123 is absent in Y124. As a follow-up to the current study, we are planning a comprehensive investigation with high counting statistics at different q -space location. If this search yields a null result, we will place a meaningful upper bound on the ratio between sharp and narrow peaks, one might be forced to conclude that the broad peak is related to oxygen defects, which are absent in Y124 – but this is speculative without a much more extensive data set, and well beyond the scope of the current study.

Reviewer #2 (Remarks to the Author):

The authors report the discovery of charge density wave (CDW) order in the stoichiometric superconductor $\text{YBa}_2\text{Cu}_4\text{O}_8$. This is believed to be a disorder free material because the absence of disorder in the chains. Surprisingly, the CDW on-set in temperature is sharper than in other cuprate systems and occurs close to T^* . I think this is an important new result that should be published in Nature Communications. The results has implications for the relationship of charge order and pseudogap in cuprates. The manuscript is very well presented, and I do not have any major comments.

Our reply: We thank the reviewer for carefully reading our manuscript and for considering it well presented and worthy of publication. In the following we answer his/her concerns one by one, and we point out the changes made in the manuscript following his/her suggestions.

I give some suggestions below:

- 1) The authors say argue that $\text{YBa}_2\text{Cu}_4\text{O}_8$ is intrinsically clean, compared to say $\text{YBa}_2\text{Cu}_3\text{O}_{6+x}$, because oxygen interstitials are not required to dope the CuO_2 planes. However, no quantitative data is provided to support this. Can the authors provide data (or refer to data) to show that $\text{YBa}_2\text{Cu}_4\text{O}_8$ is relatively clean. Data might include electronic mean free paths determined from transport or quantum oscillations.

We thank the reviewer for this suggestion. We refer to Fig. R1 above and the associated discussion. Briefly, the Cu NQR linewidth of $\text{YBa}_2\text{Cu}_4\text{O}_8$ is at least an order of magnitude narrower than those of all other underdoped cuprates including $\text{YBa}_2\text{Cu}_3\text{O}_{6+x}$, which implies an order-of-magnitude lower electronic inhomogeneity, $\Delta\rho/\rho$. The much higher inhomogeneity in Y123 and other cuprates naturally explains the broadening of the CDW transition.

Following the reviewer's suggestion, we have also checked the literature for the transport mean free path extracted from quantum oscillation measurements and found Yelland *et al.* (PRL **100**, 047003 (2008)) and Bangura *et al.* (PRL **100**, 047004 (2008)) quote values of 40 nm and 9 nm, respectively. Both values are significantly larger than the values quoted for $\text{YBa}_2\text{Cu}_3\text{O}_{6+x}$ at the same doping level, $p = 0.14$ (see e.g. Fig. R5 from La Liberté *et al.*, npj Quantum Materials **3**, 11 (2018); note that a field-induced three-dimensionally long-range ordered CDW and mean free paths of nearly 40 nm are observed for $p = 0.11$ in Y123). While this is consistent with a lower level of disorder in Y124, the values quoted by Yelland *et al.* and Bangura *et al.* are quite different, and both authors state that further work is required to determine definitive values. The transport mean free paths are therefore less meaningful than the NQR linewidths at this stage, and we therefore prefer not to discuss these values in the manuscript.

Fig. R5. Mean free path deduced from quantum oscillation measurements on $\text{YBa}_2\text{Cu}_3\text{O}_{6+x}$ (black squares), compared to the correlation lengths of 3D (open circles) and 2D (red circles) CDWs extracted from resonant x-ray diffraction (Fig. S6 of La Liberté *et al.*, npj Quantum Materials **3**, 11 (2018)).

- 2) The authors say that the CDW reflection is a ridge of scattering perpendicular to the CuO_2 planes and that measurement are made at $L \sim 1.5$ (i.e. approximately 1.5). Can the authors provide their best estimate of L (e.g. $L=1.45$) and state explicitly that measurements were made in fixed L mode (assuming this is the case). This would be helpful for others wishing to study Y124.

Our reply: We thank the reviewer for this suggestion. In our experiments, the spectrometer angle (2θ) was fixed while changing the sample angle (θ). To help other experimenters interested in studying Y124, we have added Supplementary Note 7 to the Supplement, with a figure (reproduced in Fig. R6 below) that shows the trajectory of the scans in (K,L)-space. We note that the c -axis length of Y124 is 27.25 Å, *i.e.* a roughly twice of that in Y123 (11.75 Å). For the sake of clarity, the same trajectory is also shown in the coordinates corresponding to Y123 (Note that in the previous version of the manuscript, the L-values were quoted in terms of the Y123 lattice constants; this has now been corrected.) We have also added a map of the intensity in reciprocal space to Fig. 2 (also reproduced here in Fig. R6) that explicitly demonstrates a ridge of scattering extending along L, as expected for a 2D CDW (notice that the slightly higher intensity around $L \sim 3$ in Y124 might indicate, similarly to the $L \sim 1.5$ weak maximum of the 2D CDW in Y123, the existence of a weak antiphase correlation between next nearest CuO_2 bilayers in the out-of-plane direction).

Fig. R6. (left) Accessible momentum space at the Cu-L edge absorption edge for different values of 2θ (black curves), for Y124 (top) and Y123 (bottom). The scan with the 2θ -value chosen for our experiments is marked in red. (right) Map of the intensity around the CDW reflection in reciprocal space.

3) The authors mention CDW correlation lengths in the text, e.g. "larger correlation length have been observed in Y123". Please can the authors give values of the correlation length for Y124 and values (with reference) for the larger values in Y123 they are talking about.

Our reply: In Fig. 4 and the corresponding main text, we already addressed the correlation length of the CDW in Y124 in the original version of the manuscript read by the reviewer. We have now added the correlation length in Y123 observed in high fields to the slightly modified sentence "Note that larger correlation lengths (~ 30 nm) have been observed in Y123 with $p \sim 1/8$ in high magnetic fields..."

4) The temperature dependence of the FWHM of the quasi-elastic peak is shown in the SI figure 6. I see no reason to hide this interesting plot in the SI. Please can the authors include this plot (or one of the correlation length versus temperature) in the main text.

Our reply: We thank the reviewer for this constructive suggestion. We now present the FWHM data originally shown in the SI Fig.6 as an inset into Fig. 3b in the main text.

Reviewer #3 (Remarks to the Author):

D. Betto et al. describes a Cu L-edge resonant inelastic X-ray scattering study on the stoichiometric YBa₂CuO₄08 (Y124, superconducting $T_c = 80$ K) which is free of most sources of chemical disorder. They discovered that the two-dimensional (2D) charge-order (CO) sets in sharply at $T^* = 200$ K coinciding with the onset of pseudogap (PG) suggesting a close relationship between the two phenomena. Based on the result, the authors conclude that the gradual onset of CO in other cuprates can be attributed to an inhomogeneous distribution of CO temperatures due to disorder induced by chemical substitution. The PG physics is intimately connected with superconductivity (SC) in cuprates however the origin of PG is still strongly debated. CO is known to induce Fermi surface reconstruction and partial gaps in cuprates. However, the order parameter of CO of many cuprates have gradual onset temperatures below PG onset temperature T^* . The current work on Y124 which is almost chemical disorder free sheds new light on the issue and demonstrates the coincidence between the two phenomena.

Our reply: We thank the reviewer for the critical reading of our work and for recognizing how it sheds new light on the relationship between charge order and pseudogap. We address his/her concerns below.

The data analysis and presentation of figures are clear, however, I have some concerns with regards to the part of the main text: (1) Throughout the main text, the authors repetitively highlight the sharp coincident onset of CO and PG phenomena which is the major result of the manuscript. The authors mentioned there are solid theoretical proposals on the coincidence, however, I am expecting a more in-depth discussion on the implication of the finding. For instance, on the one hand, 2D CO in the current work shows a clear competition with SC below T_c , similar to Y123 systems. If CO and PG are closely connected to each other, can authors address the mutual relationship among CO, PG, and SC at least for Y124 compound?

Our reply: We thank the reviewer for this suggestion, which has prompted us to modify and extend the last paragraph of the manuscript in such a way that it concisely describes the temperature evolution of the electron system in Y124, including the interplay between the three key phenomena mentioned by the reviewer. For convenience, we have reproduced this paragraph below (note that the references were deleted here for the sake of clarity).

We end our discussion with some remarks on the temperature evolution of electronic correlations in Y124, beginning at high temperatures where thermodynamic data indicate an additional pseudogap in the spin sector. Both these data and the paramagnon excitations shown in Fig. 2a are compatible with a scenario in which nearest-neighbor, dynamical spin-singlet correlations are established on a temperature scale set by the magnetic exchange coupling $J \sim 100$ meV. In this scenario, T^ marks the eventual condensation of these singlets into a state with a static, bond-centered charge modulation. Note that recent high-field NMR data on underdoped Y123 have also been interpreted in terms of a two-gap scenario with a high-energy spin gap that increases strongly with decreasing doping, and a low-energy, more weakly doping-dependent gap related to charge ordering. Upon further cooling below T_c , superconductivity develops in Y124 on the backdrop of a charge-ordered texture with characteristic domain size comparable to the superconducting coherence length (~ 2 nm). As charge order is weakened but not obliterated by the onset of superconductivity (Fig. 3b), both forms of electronic order are intrinsically "intertwined" at this length scale. Y124 offers a unique platform to study this intertwined ground state with minimal interference from uncontrolled disorder. Our results open up numerous perspectives for further experimental and theoretical exploration, including the response of charge order to strain and magnetic fields, and the effect of controlled disorder on the microscopic charge correlations and macroscopic electronic properties.*

We also know that in some extremely overdoped Bi2201 cuprates CO is re-entrant to the phase diagram [YY Peng et al., Nature materials, 2018] and there is no PG physics in the overdoped region, and likely the CO physics is very different to the one discovered here. Comments on the relevance of the work to the current study should be provided.

Our reply: The reviewer is right: It is indeed puzzling that the charge ordering phenomena recently reported in overdoped Bi-based and La-based cuprates (see also Li *et al.*, Phys. Rev. Lett. **131**, 116002 (2023)) do not seem to be associated with a gap or pseudogap. We refer the reviewer to a study we have just completed, which indicates that the x-ray superstructure reflections in overdoped La-based superconductors that had previously been attributed to electronic charge order are actually due to oxygen

vacancy order (<https://arxiv.org/abs/2408.06774>). If this finding can be confirmed for the Bi-based system, the puzzle will be resolved. As the current study of underdoped Y124 does not yield any insight into electronic correlations in overdoped cuprates, we prefer not to comment on this issue. However, if the reviewer insists, we will insert a corresponding remark.

Also, the discussion of the PG phenomena in the spin sector could be further elaborated. For instance, can RIXS provide some relevant information in terms of spin-singlet correlation? The discussion should be linked in connection to the relationship between PG and CO.

Our reply: We are grateful for this remark. We have streamlined and extended the discussion in the revised summary paragraph reproduced above. Specifically, we mention that the antiferromagnetic paramagnon excitations we have observed by RIXS in Y124 (Fig. 2) are indeed compatible with spin-singlet correlations. We have also inserted a reference to a recent NMR study on Y123 that discriminates between the pseudogap associated with charge ordering and an additional spin gap that opens up at high temperatures (<https://arxiv.org/abs/2402.02508>).

(2) Secondly, previous NMR studies on a lot of cuprates suggest they detect more localised or static charge order rather than the dynamic ones in X-ray scattering. From the results presented in Y124, it seems that NMR and X-ray scattering probe the same facet of the CO phenomena. It would be very useful if the authors could address along the direction.

Our reply: We agree that NMR and x-ray scattering probe the same charge ordering phenomena, albeit on different energy (and time) scales. We refer the reviewer to Wu *et al.* Nat. Comm. **6**, 6438 (2015), which is now cited in the introduction, and to Fig. R3 above, which compares x-ray and NMR results on the same Y123 compounds. The comparison is important because it demonstrates that the charge order detected by x-rays is static on the μsec time scale probed by NMR. This is now explicitly mentioned on page 2 of the manuscript, and a corresponding reference (Ref. 30) was inserted.

(3) The authors mentioned that the onset temperature of CO is subject to chemical disorder, however, that of the PG seems independent and has a universal temperature dependence regardless the specific cuprate compounds. Could authors elaborate a bit more why that is the case?

Our reply: In the revised manuscript, this point is now also addressed in a more transparent manner. As mentioned in the revised summary paragraph, thermodynamic and NMR data indicate a spin gap that forms well above room temperature and is associated with spin-singlet correlations on nearest-neighbor Cu sites. As these correlations are confined to nearest-neighbor pairs, they do not appear to be strongly affected by specific realizations of doping-induced disorder. Both the incommensurability and the domain size of the charge ordering observed by x-ray diffraction, on the other hand, manifest correlations on larger length scales comparable to the average distance between doping-induced defects, whose nature and distribution depends on the specific material under study.

The authors have found that the CO domain size is largely independent of the distribution of oxygen defects unlike the onset temperature of CO. Again, it would be very beneficial to general audience by providing a qualitative explanation.

The small domain size of charge order in Y124 is one of the key findings of our study, which is highlighted more clearly in the revised version of our manuscript (page 4-5). In view of the homogeneity of the electron system of Y124, our results indicate that the domain size is primarily controlled by the intrinsic properties of the electron system in the CuO_2 planes, rather than materials-specific disorder. In addition to general theoretical arguments on the instability of incommensurate CDWs to domain formation (following the classical Imry-Ma argument), specific theoretical work on the cuprates indeed predicts that uniaxial charge order is high susceptible to transverse fluctuations and to topological defects. The near-degeneracy of multiple forms of order in the strongly correlated electron system further lowers the energy cost of domain walls. As emphasized in the last paragraph of the manuscript, our results establish Y124 as a unique platform to study this physics with minimal interference from uncontrolled disorder.

(4) Extrapolating the current major result, that is, the coinciding onset temperatures of PG and CO, to the conclusion of the gradual onset of CO in many other cuprates is due to the disorder induced by chemical substitution is very indirect and a bit of stretch. I would be much satisfied if the current work comprises a

parallel study on Y124 with the introduction of disorder, either by chemical doping or by electron irradiation. If the study could demonstrate the shift of the sharp onset temperature CO to a gradual one and differentiate from that of PG, it would be a more sophisticated study to convince the readers about the conclusion.

Our reply: We refer the reviewer to Figs. R1 and R2 and the discussion on page 3-4 of this Rebuttal. There is a clear and obvious correlation between the exceptional homogeneity of the electron system in Y124 (Fig. R1) and the exceptionally sharp onset of charge order (Fig. R2). While there are other possible explanations of the gradual onset of the charge order parameter in other cuprates (which are now briefly discussed on page 4 of the manuscript), inhomogeneous broadening is by far the most straightforward explanation in the light of our results on a homogeneous system. Nonetheless, in view of the reviewer's skepticism, we have softened the corresponding language in the abstract and in the discussion.

Following up on our findings on the clean system, we are planning further studies of Y124 with controlled amounts of disorder introduced by Ca and La substitution. We have already started a synthesis and characterization effort, but it will take at least a year to complete this effort, to secure synchrotron beamtime, and to perform resonant x-ray scattering experiments on the impact of disorder on the charge-ordered state. We hope and expect, however, that the reviewer's concerns have been adequately addressed, and that he/she will share our conviction that the current discovery paper on a system that has played a prominent role in research on high-temperature superconductivity merits publication in Nature Communications.

(5) Finally, a much minor point about the explanation of the CO. A lot of text goes to explain the resonance effect of CO in CuO₂ planes instead of CuO chains. In my opinion the text could be easily relocated to Supplementary Information as this has been confirmed by previous works extensively.

Our reply: As pointed out, the resonant profile of the charge-ordering superstructure reflections has been repeatedly studied in Y123, and yet this is the first observation in the double-chain compound Y124. Since the peculiar chain structure is one of the greatest differences between Y123 and Y124, we believe that the resonant profile showing that the charge order only affects the CuO₂ planes is important, and we would like to keep it in the main text. However, we agree with the reviewer that it does not merit a standalone figure. We have therefore integrated this profile as one of four panels into the revised Figure 2.

Reply to Reviewers for Nature Communications manuscript NCOMMS-23-60389A

Coincident onset of charge order and pseudogap in a homogeneous high-temperature superconductor

D. Betto*, S. Nakata* *et al.*

REVIEWER COMMENTS

Reviewer #1 (Remarks to the Author):

In their original manuscript, Betto et al. convincingly demonstrated the coincidence of charge density wave (CDW) onset and pseudogap temperature in $\text{YBa}_2\text{Cu}_4\text{O}_8$ (Y124) using resonant inelastic X-ray scattering (RIXS). I previously noted that this finding is significant enough to merit publication in Nature Communications. However, I raised two main concerns.

First, regarding the presentation of the data, I observed that the authors did not cite or properly contextualize recent literature presenting analogous results. Second, concerning the interpretation of the data, I did not find the attribution of this coincidence primarily to the minimal disorder and crystallographic perfection of Y124 compared to other cuprates to be very convincing.

In the revised version, I appreciated the authors' shift in focus, as indicated already by the title, which emphasizes the electronic homogeneity of Y124. This perspective highlights the potential for more precise examination of the coincidence of CDW and pseudogap onset using this "platform", with less uncertainty than in any other cuprate compounds. I agree with the authors that this argument significantly strengthens the paper.

Regarding my first concern, the authors have now appropriately cited a relevant work as Ref. 14, which presents similar findings. In their response, they acknowledged the importance of this work, stating that the comprehensive results emerging from these experiments underline the necessity of studying the effects of controlled disorder and inhomogeneity on orders such as CDW and pseudogap, thereby enhancing their entwining. I believe this point is important and suggest that the authors give the proper relevance to these findings not only citing the reference in question, but discussing the relevance of this previous work in the context of their findings.

Based on these considerations, and provided that these minor changes are made, I strongly recommend the paper for publication in Nature Communications.

As an additional comment regarding Figure 3b, I remain unconvinced by the authors' arguments. With the advances in experimental techniques, the RIXS setup currently used by the authors to measure Y124 exhibits significantly improved sensitivity and signal-to-noise ratio compared to the equipment available approximately 15 years ago, on which the gray curve, reproduced from Ref. [35], is based. However, I would like to emphasize that Figure 3b does not impact the main findings or interpretations of the paper.

Our reply: We thank the Reviewer for recognizing and appreciating our efforts to improve the manuscript and the shift in focus to better highlight the nature and consequences of our findings, and we thank him/her for recommending publication of the paper in Nature Communications.

In the last round of reviewing, we had complied with Reviewer #1's request to cite the paper by Wahlberg et al., (now Ref. [14]). This is appropriate, in our view, because Ref. [14] reports one Y123 thin-film sample where the stated charge-density-wave transition temperature coincides with the pseudogap onset temperature T^* within the combined experimental error. We had digitized the resonant x-ray scattering data from this publication together with Y123 x-ray data from other publications (Supplementary Note 6), so that the readers of Nature Communications can assess them directly. Following the Reviewer's request for additional discussion of Ref. [14] along the lines of our response to the first round of review, we have copied the pertinent paragraph of our response letter (with slight editorial modifications) into Supplementary Note 6. This paragraph discusses merits and limitations of measurements on thin films. We end this paragraph with a sentence that mirrors the Reviewer's statement above:

The results reported for Y123 films reported in Ref. [14] are thus complementary to our study of Y124 bulk crystals, and call for further investigations of the effects of controlled disorder and inhomogeneity on the CDW and the pseudogap.

We also agree with the Reviewer that the current RIXS setup has improved sensitivity compared to the situation 15 years ago, and that this sensitivity enhancement does not affect the overall T-dependence of the charge order intensity in Y123 and the qualitative difference with the one in Y124. As pointed out by the Reviewer, the main findings and interpretations of the paper are not impacted by these technical issues.

Reviewer #2 (Remarks to the Author):

I am happy that the authors have addressed my comments and recommend publication subject to addressing issues raised by the other referees.

Our reply: We thank the Reviewer for taking the time to review our work one more time, for recognizing our efforts to address his/her comments, and for recommending publication in Nature Communications.

Reviewer #3 (Remarks to the Author):

I am happy that the authors have addressed my comments and recommend publication subject to addressing issues raised by the other referees. It is great to see that D. Betto et al. improved the manuscript by considering the comments/suggestions from the referees. However, I still have concerns on the revised version.

(1) PG temperature T^*

In Fig.3c shows the temperature dependence of the difference in O2 and O3 spin susceptibilities obtained from Knight shifts, 17K2,c - 17K3,c in NMR measurements on Y124 from Ref. [24]. The authors state that the sharp anomaly at 200 K serves as proxies of the pseudogap. The authors also highlight "this temperature coincides with $T^* \sim 200$ K, where the opening of a pseudogap with amplitude ~ 50 meV has been determined directly by infrared [37] and Raman [36] spectroscopy." Upon further study, Ref. 24 stated (in the caption of Fig.2)

that the difference in O2 and O3 spin susceptibilities obtained from 17K2,c - 17K3,c shows the abrupt onset of nematic splitting at 200 K within the pseudogap state. In the 1st paragraph in page 4 of Ref.24, they also mentioned that the pseudogap extends at least to well above 400 K, and the nematic phase transition occurs within a preexisting pseudogap state that extends far above. In fact, one of the key results of Ref.24 (as stated in their Abstract) is the extensive field-dependent specific heat studies on YBa2Cu4O8 up to 400 K and showed that the pseudogap never closes, remaining open to at least 400 K where T^* is typically presumed to be ~ 150 K. Given that the comparison of the temperature-dependence between the current CDW study and the NMR Knight shift data (Ref.24) is critical to the key message of the work, I would like to invite the Authors to address the discrepancy. Broadly speaking, if there are challenges to the usual understanding of the pseudogap temperature at least for the same cuprate YBa2Cu4O8, the key message about the coincidence of the CDW characteristic temperature with respect to that of pseudogap becomes less straightforward and valuable.

Our reply: We thank the Reviewer for the positive comments on the revised version of the manuscript, and for their conditional acceptance recommendation. We highly appreciate their remarks, with which we completely agree on substance, and we hope and expect that the presentation and discussion in the new version of the manuscript properly reflect this common understanding.

As the Reviewer correctly points out, a complete review of the literature on Y124 reveals two separate pseudogap phenomena:

1. a pseudogap that opens at $T^* = 200$ K, as directly evidenced by infrared spectroscopy and electronic Raman scattering, and indirectly by the behavior of the oxygen NMR Knight shifts and anomalies in other observables. It is this pseudogap and the associated transport anomalies that has captured much of the research field's attention over the past few years, and is hence commonly referred to as "THE pseudogap". We are certain that our association of this pseudogap with charge order will be widely seen as a key advance not only by specialists in high-temperature superconductivity, but also across other fields of research that have been inspired by the phenomenology of the cuprates.
2. a second pseudogap that is not directly seen in spectroscopy, but has been uncovered via a "missing entropy" in a comprehensive survey of thermodynamic data on multiple cuprate compounds including Y124. This second pseudogap is temperature independent – at least up to 400 K, the temperature range covered by Ref. [24] and most other studies of this kind – and is hence not associated with any temperature dependent transport anomalies. It has therefore generated far less attention than "pseudogap 1" mentioned above. The semantic confusion arises from the fact that Ref. [24] (quoted by the Reviewer) and some other articles in the literature refer to this pseudogap 2 as "THE pseudogap", and to T^* as a phenomenon within "THE pseudogap state".

In the new version of our manuscript, we explicitly address the distinction between both pseudogap phenomena in the penultimate paragraph. Both phenomena fit naturally into a scenario in which "pseudogap 2" is attributed to the formation of dynamical spin-singlets via the nearest-neighbor exchange interaction (whose large magnitude, $J \sim 1500$ K, explains its high-temperature onset), and the formation of "pseudogap 1" at T^* is associated with the condensation of these singlets into bond-centered charge stripes.

We have also critically examined our manuscript and found two instances in which we may have inadvertently contributed to the semantic confusion by using the common (but inappropriate) phrase “THE pseudogap” for “pseudogap 1”. We have changed the wording to avoid any misunderstanding.

We thank the Reviewer for forcing us to improve our discussion. Of course, these changes should in no way detract from the importance of our finding that pseudogap 1, which has generated an enormous amount of debate in the field, is associated with charge order.

In the previous report, I had a question concerning the sensitivity of the pseudogap temperature to the chemical disorder. The authors suggested that both thermodynamic and NMR data indicate a spin gap forms well above room temperature which associates with spin-singlet correlation. I am confused that the authors seem to allude the pseudogap to the spin gap which clearly exceeds $T^* \sim 200$ K rather than the charge degree of freedom. More specifically, in 2nd paragraph, p4 of the revised manuscript, the authors state that “the pseudogap onset temperature is generally ~ 200 K for cuprates with $p \sim 0.14$ ”. By searching the literatures, LSCO cuprates show a characteristic $T^* \sim 150$ K rather than 200 K.

Our reply: Our data do not speak directly to the origin of “pseudogap 2”, which opens up at very high temperature. (In fact, Ref. [24] and other work on Y124 we are aware of only report lower bounds on this temperature.) We have added two references to NMR work on other cuprates supporting the assignment of “pseudogap 2” to the spin sector. In addition, our observation of paramagnons with characteristic energies of ~ 300 meV (Fig. 2a) demonstrates that the nearest-neighbor superexchange coupling $J \sim 1500$ K is active in Y124, and there is a lot of theoretical work demonstrating that this interaction generates a spin-pseudogap in correlated metals.

The referee is correct to say that the reported values of T^* for the LSCO family of cuprates are lower than those of other families (see the reviews in Refs. [1] and [45]), perhaps as a consequence of the higher influence of disorder. We have reworded two pertinent sentences as follows:

p.2 *Prior experiments have shown that the pseudogap in Y124 opens up below $T^* \sim 200$ K, in line with **most** other cuprates at this doping level*

p. 4 *However, the pseudogap onset temperature is ~ 200 K for **most** cuprates with $p \sim 0.14$*
...

(2) CO in overdoped cuprates

I still think the inclusion of a brief clarification of the CDW study in YBa₂Cu₄O₈ with respect to the CO in overdoped cuprates would be useful to the general audience. At least this shows the complex multiple facets of the charge correlation.

Our reply: Following the Reviewer’s advice, we have (somewhat reluctantly) added the following sentence at the very end of the manuscript:

Finally, we note that our results do not speak directly to recently reported observations of charge order in overdoped cuprates, where a pseudogap is not present but different factors such as a van-Hove singularity or oxygen-vacancy order may be relevant.

(3) CO domain size

In the reply, the authors stated that the domain size is primarily controlled by the intrinsic properties of the electron system in the CuO₂ planes rather than materials-specific disorder. However, in the 1st paragraph, p2 of the revised manuscript, the authors state that “Moreover, general theoretical arguments imply that 2D systems with incommensurate charge order break up into finite-sized domains in response to arbitrary low levels of disorder [20]”. The authors should at least reconcile the statement by them with the quotation about the theoretical proposals. It would be reasonable to think the disorder / impurities break up the large domain into smaller finite-sized domains as commonly seen in metal-insulator transition systems and multiferroics.

Our reply: Our work demonstrates that doping-induced disorder affects charge ordering primarily via its influence on the homogeneity of the electron system. Our findings on Y124 further indicate that its effect on the domain size is comparatively minor. One factor that is likely to be relevant for models of this behavior is the anomalously low energy cost of defects in incommensurate charge-ordered states in two spatial dimensions that has already been reported in theoretical work. The detailed mechanisms underlying the formation and pinning of these electronic defects – including the possible relevance of residual oxygen vacancies/interstitials, chemical impurities, and microstructural features such as dislocations – remain to be elucidated by future research.

(4) About Fig.2d

The authors clarified that RIXS experiment was done in a fixed $2q$ geometry. The Q -dependence follows the single curve marked in red in Fig. R6. If so, how to construct the 2D (K , L) map from the existing data?

Our reply: As discussed in the first reply, in our experiments, the spectrometer angle (2θ) was fixed while changing the sample angle (θ). This allows us to track and follow a single curve like the red one in Fig. R6. However to collect and construct our 2D KL map we had of course to change the spectrometer angle 2θ and indeed we measured several θ -scans, i.e. several curves like the blue ones in Fig. R6, each at a different 2θ , i.e. with a different maximum momentum and thus around a different starting L value. The red curve in R6 simply represents the one with the highest reachable momentum at Cu L₃ edge with the ERIXS spectrometer and also the geometry used to collect the θ -scans in Figs. 2a,b and Fig. 3a. We have corrected the corresponding caption of Supplementary Figure 8 to make it clearer.

Additional revision

We have made one additional revision that was not directly prompted by the Reviewers, but rather by a discussion at a recent conference, which cast some doubt on the uniqueness of the analysis and conclusions of Ref. [25]. Specifically, the conclusion that the resistivity reported in this paper provides evidence for pseudogap formation at T^* may be model specific and possibly subject to confirmation bias. We have therefore decided not to show the resistivity data on Ref. [25] in Fig. 3, and slightly revised the discussion of Fig. 3 in the text on p. 4. The revised Fig. 3 is shown below. The spectroscopic data displayed in panel c stand on their own.

Fig. R1: Revised Fig. 3. In particular Fig.3c was revised.

It is great to see that D. Betto et al. improved the manuscript by considering the comments/suggestions from the referees. However, I still have concerns on the revised version.

(1) PG temperature T^*

In Fig.3c shows the temperature dependence of the difference in O2 and O3 spin susceptibilities obtained from Knight shifts, $17K_{2,c} - 17K_{3,c}$ in NMR measurements on Y124 from Ref. [24]. The authors state that the sharp anomaly at 200 K serves as proxies of the pseudogap. The authors also highlight “this temperature coincides with $T^* \sim 200$ K, where the opening of a pseudogap with amplitude ~ 50 meV has been determined directly by infrared [37] and Raman [36] spectroscopy.”

Upon further study, Ref. 24 stated (in the caption of Fig.2) that the difference in O2 and O3 spin susceptibilities obtained from $17K_{2,c} - 17K_{3,c}$ shows the abrupt onset of nematic splitting at 200 K *within* the pseudogap state. In the 1st paragraph in page 4 of Ref.24, they also mentioned that the pseudogap extends at least to well above 400 K, and the nematic phase transition occurs *within* a preexisting pseudogap state that extends far above. In fact, one of the key results of Ref.24 (as stated in their Abstract) is the extensive field-dependent specific heat studies on YBa₂Cu₄O₈ up to 400 K and showed that the pseudogap never closes, remaining open to at least 400 K where T^* is typically presumed to be ~ 150 K.

Given that the comparison of the temperature-dependence between the current CDW study and the NMR Knight shift data (Ref.24) is critical to the key message of the work, I would like to invite the Authors to address the discrepancy. Broadly speaking, if there are challenges to the usual understanding of the pseudogap temperature at least for the same cuprate YBa₂Cu₄O₈, the key message about the coincidence of the CDW characteristic temperature with respect to that of pseudogap becomes less straightforward and valuable.

In the previous report, I had a question concerning the sensitivity of the pseudogap temperature to the chemical disorder. The authors suggested that both thermodynamic and NMR data indicate a spin gap forms well above room temperature which associates with spin-singlet correlation. I am confused that the authors seem to allude the pseudogap to the spin gap which clearly exceeds $T^* \sim 200$ K rather than the charge degree of freedom.

More specifically, in 2nd paragraph, p4 of the revised manuscript, the authors state that “the pseudogap onset temperature is generally ~ 200 K for cuprates with $p \sim 0.14$ ”. By searching the literatures, LSCO cuprates show a characteristic $T^* \sim 150$ K rather than 200 K.

(2) CO in overdoped cuprates

I still think the inclusion of a brief clarification of the CDW study in YBa₂Cu₄O₈ with respect to the CO in overdoped cuprates would be useful to the general audience. At least this shows the complex multiple facets of the charge correlation.

(3) CO domain size

In the reply, the authors stated that the domain size is primarily controlled by the intrinsic properties of the electron system in the CuO₂ planes rather than materials-specific disorder. However, in the 1st paragraph, p2 of the revised manuscript, the authors state that “Moreover, general theoretical arguments imply that 2D systems with incommensurate charge order break up into finite-sized domains in response to arbitrary low levels of disorder [20]”. The authors should at least reconcile the statement by them with the quotation about the

theoretical proposals. It would be reasonable to think the disorder / impurities break up the large domain into smaller finite-sized domains as commonly seen in metal-insulator transition systems and multiferroics.

(4) About Fig.2d

The authors clarified that RIXS experiment was done in a fixed 2θ geometry. The Q-dependence follows the single curve marked in red in Fig. R6. If so, how to construct the 2D (K, L) map from the existing data?